# Parental Knowledge, Awareness, and Attitudes Toward Seasonal Influenza Vaccination in Al-Madinah, Saudi Arabia: A Cross-Sectional Study

**DOI:** 10.3390/ijerph22111704

**Published:** 2025-11-11

**Authors:** Abdulsalam Alawfi, Muhammad Tobaiqi, Osama Algrigri, Amal H. Aljohani, Amal Mohammed Q. Surrati, Bandar Albaradi, Amer Alshengeti

**Affiliations:** 1Department of Pediatrics, College of Medicine, Taibah University, Al-Madinah 41491, Saudi Arabia; ogrigri@taibahu.edu.sa (O.A.); ahjohani@taibahu.edu.sa (A.H.A.); aalshengeti@dal.ca (A.A.); 2Department of Pediatrics, King Faisal Specialist Hospital and Research Center, Al-Madinah 42523, Saudi Arabia; 3Department of Family and Community Medicine and Medical Education, College of Medicine, Taibah University, Al-Madinah 41491, Saudi Arabia; mtobegi@taibahu.edu.sa (M.T.); asurrati@taibahu.edu.sa (A.M.Q.S.); 4Department of Pediatrics Infectious Diseases, King Fahad Specialist Hospital, Dammam 32222, Saudi Arabia; bandar.baradi@hotmail.com; 5Department of Infection Prevention and Control, Prince Mohammad Bin Abdulaziz Hospital, National Guard Health Affairs, Al-Madinah 41491, Saudi Arabia

**Keywords:** influenza, vaccine, children, parent

## Abstract

The World Health Organization and the Centers for Disease Control and Prevention recommend seasonal influenza vaccination for all individuals aged 6 months and older. Despite high national immunization rates, the influenza vaccination coverage among Saudi children remains unclear. Parental knowledge and attitudes significantly impact children’s vaccination rates. **Purpose**: This study aims to evaluate parental knowledge, awareness, and attitudes regarding influenza vaccination and identify barriers to vaccination uptake among children in Al-Madinah City, Saudi Arabia. **Methods**: The population includes parents having children aged 6 months to 14 years. A cross-sectional survey utilizing a 33-item validated questionnaire was conducted to evaluate parental awareness, knowledge, and attitudes toward the influenza vaccine. Inferential statistics were employed to evaluate demographic factors influencing parental knowledge and attitudes toward vaccination. **Results**: This study surveyed 407 parents from Al-Madinah, focusing on their awareness, knowledge, and attitudes towards seasonal influenza vaccination. The sample was primarily Saudi (86.7%), with a mean age of 34 years. Most parents (95.6%) were aware of the vaccine, primarily through media and campaigns. Despite this, only 44.5% had vaccinated themselves or their children, citing perceptions of influenza as mild, vaccine ineffectiveness, and availability issues as primary reasons for non-vaccination. Knowledge about influenza varied, with most parents aware of its contagiousness (64.4%) and symptoms, but misconceptions persisted, such as believing the vaccine could cause the flu. Parental attitudes towards vaccination were mostly positive, with high trust in health information sources and a mean attitude score of 22.48 out of 35. Positive attitudes were correlated with better knowledge and more frequent infection control practices. Age, education, and medical profession status significantly influenced knowledge, while vaccine attitudes were most favorable among those vaccinated (*p* < 0.001). **Conclusions**: Most parents in Al-Madinah recognize the importance of vaccination; however, misconceptions about vaccine safety, perceived low need, and barriers such as vaccine availability persist. Sociodemographic factors, including education, income, and profession, are linked to better knowledge and more positive attitudes toward vaccination.

## 1. Introduction

Influenza viruses induce a globally significant, contagious respiratory illness characterized by diverse systemic manifestations. Vulnerable populations, including young children, the elderly, and those with comorbidities, experience a heightened risk of severe outcomes ranging from mild febrile illness and cough to critical complications such as respiratory failure and mortality, particularly within the pediatric demographic [1,2,3]. Global influenza incidence is estimated by the World Health Organization (WHO) at approximately one billion cases annually, resulting in 3–5 million severe cases and 290,000–650,000 deaths attributable to respiratory complications. This disease burden is further compounded by significant economic costs associated with direct healthcare expenditures and indirect losses due to reduced productivity from absenteeism in work and educational settings [4,5].

Pediatric populations exhibit heightened susceptibility to severe influenza complications, encompassing pneumonia, hospitalization, and neurological sequelae such as convulsions and encephalopathy, while also serving as key vectors for household transmission. Nonetheless, seasonal influenza vaccines (SIVs) offer effective prophylaxis, demonstrably reducing the incidence of influenza-related hospitalizations and mortality within this demographic [6,7].

Global health organizations, including the WHO and CDC, recommend SIVs for individuals 6 months and older, prioritizing high-risk groups and those in contact with them. Specifically, children aged 6 months to 5 years (WHO) and all individuals 6 months and older (CDC) are targeted. Primary vaccination for children aged 6 months to 8 years requires two doses administered at least four weeks apart, followed by a single annual dose thereafter at the beginning of each influenza season [8,9].

Influenza outbreak management in Saudi Arabia is complicated by the Hajj pilgrimage, which often coincides with peak influenza transmission periods. While the US Advisory Committee on Immunization Practices (ACIP) recommends annual influenza vaccination for all individuals over 6 months of age (barring contraindications), and Saudi Arabia reports national immunization coverage exceeding 90%, the specific vaccination rate among Saudi children remains undetermined, despite endorsements from the Ministry of Health and the Saudi Thoracic Society [10,11]. The previous literature suggests that several factors significantly impact parental decisions regarding influenza vaccination for their children [12]. Notably, parental education level positively correlates with favorable attitudes toward immunization [12]. Furthermore, external influences, such as physician recommendations and media campaigns across various platforms (newspapers, television, and internet), have been demonstrated to shape parental awareness and attitudes towards influenza vaccination [13,14,15,16].

Limited research in Saudi Arabia has explored the factors influencing the administration of influenza vaccines to children [12,17,18]. This study aimed to evaluate parental knowledge and attitudinal determinants regarding influenza vaccination, emphasizing identifying barriers to pediatric vaccine uptake. The research objective focused on elucidating decision-making factors that influence parental vaccine acceptance, informing evidence-based interventions to enhance seasonal influenza vaccination coverage in the pediatric population.

## 2. Materials and Methods

### 2.1. Study Design and Setting

This cross-sectional study was conducted at primary healthcare centers in Al-Madinah city, targeting parents residing in the area with at least one child aged 6 months to 14 years. The study followed the Strengthening the Reporting of Observational Studies in Epidemiology (STROBE) guidelines [19].

#### 2.1.1. Inclusion and Exclusion Criteria

Parents with children older than six months who resided in Al-Madinah were included. Parents were excluded if they did not have children, had children under six months of age, or could not communicate in Arabic or English.

#### 2.1.2. Outcomes

The primary outcomes were:Parents’ knowledge scores regarding childhood influenza vaccination;Parents’ attitudes toward childhood influenza vaccination;Parents’ practices regarding childhood influenza vaccination.

The secondary outcome was parents’ attitudes toward influenza vaccination during the COVID-19 pandemic.

### 2.2. Participants and Recruitment

Participants were recruited from primary healthcare centers. The centers were distributed across different regions of Al-Madinah to ensure representation from different communities, where interviews were conducted. When parents agreed to complete the interview, consent for participation was obtained.

### 2.3. Data Collection

The data collection instrument comprised a validated 33-item survey adapted from Pakistani research [20], encompassing demographic variables (*n* = 8), parental knowledge and attitudes toward pediatric influenza vaccination (*n* = 22), and COVID-19-specific vaccination attitudes (*n* = 3). The survey was translated from English to Arabic language to facilitate the interview by the data collector team members who are bilingual. Ten interviews were conducted before the start of the study to ensure the clarity of the content. Knowledge assessment utilized a binary scoring system (poor: 0–5; good: >5), while attitudinal measures employed a categorical scale (negative: 7–21; positive: 22–35). Survey validation included pilot testing with 20 participants, with subsequent modifications based on feedback. The interview was verbal, with the data collector filling out the data form at the time of the interview.

### 2.4. Sample Size

Based on the national census, the population of children aged 0 to 18 years in Al-Madinah was 495,119 [21]. A sample size of 384 participants was calculated with a 95% confidence level and a 5% margin of error, which was adjusted to 400 to account for incomplete responses. The required sample size was calculated using Epi Info software (Version 7.1.5. Centre for Disease Control and Prevention, Atlanta, GA, USA). Patients were selected for the study through convenience sampling.

### 2.5. Statistical Analysis

The mean and standard deviation were used to describe continuous variables and the frequencies and percentages for the categorically measured variables. The statistical Kolmogrove–Smirnove test of statistical normality and a histogram were used to assess the normality of the continuous variables, and Levene’s test was applied to assess the equality of variance statistical assumption. Cronbach’s alpha test was used to assess the reliability of the parental attitude to the flu vaccine by a seven-item questionnaire.

A total score for people’s attitudes was computed via adding up their responses to the seven indicators measuring their flu vaccine attitudes, yielding a total score between 7 and 35 points, and a total knowledge score was measured via adding the correctly answered knowledge questions, measuring peoples general awareness of the flu disease and of signs and symptoms, yielding a total knowledge score between 0 and 16 points. Each correctly answered question was rewarded with one point.

The unpaired samples *t*-test was used to assess the statistical significance of the mean difference for the continuous measured variables across the levels of binary categorically measured variables, and the one-way ANOVA test used to compare metric variables across the levels of the categorical variables of >2 groups. An adjusted *t*-test and one-way ANOVA tests were used when the statistical assumptions of these tests were violated (e.g., when unequal variances were noted).

The multivariable standard linear regression analysis was applied to assess the predictors for parents’ attitudes and knowledge scores on pediatric flu vaccines and disease. The collinearity diagnostics were evaluated with the VIF and Tolerance indices and the normality assumptions for standard linear regression were assessed with the PP plots. The association between the predictors and the analyzed outcomes in the linear regression analysis was expressed as a beta coefficient with its associated 95% confidence intervals. The SPSS IBM version 22 for Windows (IBM Corp., Armonk, NY, USA) was used for the statistical data analysis and the alpha significance level was considered at the 0.050 level.

## 3. Results

### 3.1. Parents’ Sociodemographic Characteristics

Our study recruited 407 parents from residents in Al-Madinah. The sample included 56.5% females and 43.5% males. Regarding age distribution, the majority were 25–34 years old (40%), followed by 35–44 years (34.2%), 18–24 years (15%), 45–54 years (7.1%), and ≥55 years (3.7%). Most participants were Saudi nationals (86.7%), while 13.3% were non-Saudi. Educational attainment showed that 51.1% held a college/university degree, 30.2% had completed high school, 10.1% had less than a high school education, and 8.6% had higher degrees. Regarding household income, 32.4% chose not to disclose, while 24.6% earned SAR 2000–5999, 23.1% earned SAR 6000–9999, 15.5% earned SAR 10,000–15,000, and 4.4% earned above SAR 15,000. The majority (80.6%) were not medical professionals, and 19.4% were. The mean number of children per household was 2.40 (SD = 1.5) (Table 1).

### 3.2. Parents’ Awareness of SIV

The analysis of parental awareness and behavior regarding seasonal influenza vaccination revealed that 95.6% had prior knowledge of the vaccine, primarily through media (40.1%) and awareness campaigns (19.8%). Among parents, 59.7% reported receiving the influenza vaccine themselves, while 72.5% stated their children were up to date with the national vaccination program. However, only 44.5% indicated that they or their child had received the flu vaccine, with 40% reporting no prior vaccination and 15.5% unsure. The primary reasons for not vaccinating included the perception that influenza is simple or that there is no need to vaccinate (35.9%), followed by beliefs about ineffectiveness (25.8%) and lack of availability (23.4%) (Table 2).

### 3.3. Parents’ Knowledge About SIV

Parental knowledge of the seasonal influenza vaccination showed that 77.1% recognized influenza as common and 64.4% understood its contagious nature. Most parents knew that it spreads through coughing and sneezing (76.4%), but fewer were aware of its potential to cause hospitalization or death (47.7%). Vaccine recommendations were moderately understood (60.2%) and 45.7% knew that it reduces severity. Misconceptions persisted, with 76.9% mistakenly believing the vaccine causes flu. Knowledge of vaccine frequency was high (70.3%), but only 22.4% knew that antibiotics do not work against influenza. Flu symptoms such as fever (84.3%) and runny nose (65.1%) were widely recognized, with lower awareness of other symptoms. The mean knowledge score for influenza symptoms was 2.84 (SD = 1.60) out of 6, and for influenza disease, it was 5.59 (SD = 2.30) out of 10. The overall knowledge score, combining both symptoms and disease, averaged 8.43 (SD = 3.10) out of 16 (Table 3).

Knowledge scores (out of 16) showed no significant difference between males (8.24, SD = 3.14) and females (8.57, SD = 3.04) (*p* = 0.297). Age significantly influenced scores, with the youngest group (18–24) scoring lowest (7.72, SD = 3.16) and those aged ≥55 scoring highest (5.67, SD = 3.50) (*p* = 0.001). Education level, income, and medical profession status significantly impacted scores (*p* < 0.001). Greater knowledge was observed in those aware of the vaccine, those with vaccinated parents, and parents of children with up-to-date vaccinations. No significant differences were found for the COVID-19 infection or vaccine willingness (*p* = 0.253, *p* = 0.331) (Table 4).

### 3.4. Parents’ Attitude Towards SIV

Parental attitudes towards the seasonal influenza vaccine for children were largely positive. The majority (95.8%) agreed that vaccinations are essential for children’s health, with a high mean score of 3.53 (SD = 0.60). Most parents also viewed the vaccine as necessary (77.2%, mean = 3.10, SD = 0.78), safe (81.1%, mean = 3.11, SD = 0.72), and effective (75.7%, mean = 3.01, SD = 0.76). However, 31% disagreed that influenza can cause serious complications (mean = 2.90, SD = 0.92), highlighting an awareness gap. Trust in health information was high, with 95.6% trusting the Saudi Ministry of Health and 94.4% trusting their doctor (mean scores of 3.51, SD = 0.60, and 3.47, SD = 0.61, respectively). Most parents (77.9%) practiced infection control measures, with 50.6% doing so frequently. COVID-19 impacted 51.6% of families and 68.1% were willing to receive the influenza vaccine. Overall, parents had a positive attitude, with a mean score of 22.48 (SD = 3.39) out of 35 (Table 5).

There was no significant difference in attitude scores (out of 35) between males (22.13, SD = 3.50) and females (22.74, SD = 3.30) (*p* = 0.071). Attitudes were not significantly influenced by age, nationality, or household income, though higher education was associated with slightly higher scores. Parents who received the flu vaccine (23.04, SD = 3.27) had more positive attitudes than those who did not (21.65, SD = 3.42). Positive attitudes were also seen among parents with children who were up to date on vaccinations and those willing to receive the flu vaccine this year (*p* < 0.001). COVID-19 infection status had no significant effect (*p* = 0.116) (Table 6).

### 3.5. Multivriate Regression Analysis for Parents’ Attitudes and Knowledge Regarding SIV and Flu Disease

The multiple linear regression analysis showed that several predictors were associated with knowledge of the SIV and flu disease. Participants aged 55 years or older had significantly lower knowledge scores compared with younger age groups (β = −0.372, 95% CI, 0.684, −0.060, *p* = 0.020). In contrast, higher household income (SAR ≥10,000/month) was associated with greater knowledge (95% CI, 0.16–0.66, *p* = 0.001).

Regarding the attitude toward the SIV, number of children, having received the influenza vaccine as a parent, mean knowledge of flu symptoms and signs, and mean general knowledge of the flu disease were statistically significant. Table 7 (a) and (b) show the details of the multivariate analysis.

## 4. Discussion

Our study surveyed 407 parents in Al-Madinah, with a majority of females (56.5%) and a predominant age range of 25–44 years. Most participants were Saudi (86.7%) and over half held a college degree (51.1%). Awareness of the influenza vaccine was high (95.6%), primarily through media (40.1%) and campaigns (19.8%). While 59.7% of parents had received the vaccine, only 44.5% had vaccinated their children. Barriers to vaccination included perceived low need (35.9%), ineffectiveness (25.8%), and availability issues (23.4%). Parental knowledge about influenza was moderate, with misconceptions such as 76.9% wrongly believing the vaccine could cause illness. Attitudes were mostly positive, with 95.8% agreeing vaccines are vital for children’s health. Trust in health authorities was high and 68.1% were willing to vaccinate this year. Socioeconomic factors influenced knowledge and attitudes, with those having received higher education, of higher income, or medical professionals showing better awareness. No significant differences were found based on nationality, age, or COVID-19 infection history.

Parents play a significant role in their children’s decision to receive the influenza vaccine, with their understanding and attitudes toward vaccination being crucial factors [22,23]. Vaccine hesitancy is influenced by misconceptions about safety and effectiveness, concerns about side effects, and lack of awareness about the vaccine’s importance [22,23]. Additionally, cultural beliefs, socioeconomic status, healthcare access, and information sources also affect parental vaccination decisions [24,25,26].

This study examined vaccination coverage and awareness among Saudi parents, finding high rates of childhood vaccination per the national program and significant parental awareness of the seasonal influenza vaccine. These results align with prior research reporting influenza vaccine awareness levels of around 85.5–89% in Saudi populations, though the comparison studies varied in their target demographics. Notably, Alabbad et al. included three cohorts—parents, adult patients, and healthcare workers—while both Alolayan et al. and this study specifically targeted parents [12,17].

Seasonal influenza vaccine coverage among children varies considerably internationally, ranging from 6.6% in Pakistan to 57% in the United States [20,27,28,29]. This study found that 95% of Saudi parents were aware of the seasonal influenza vaccine, with 59.7% of parents and 44.5% of children vaccinated. However, a previous study in Riyadh reported lower rates, with just over a quarter of parents and 18% of children vaccinated against seasonal influenza [18]. Regional variations may account for the differences in seasonal influenza vaccine coverage observed between this study conducted in Al-Madina and the previous Riyadh-based study. Factors such as public health initiatives and community engagement in Al-Madina may have contributed to the higher vaccination rates that were reported. Importantly, awareness of vaccine availability and parental vaccination practices remain key predictors in improving influenza vaccine coverage among children.

Our study identified misinformation as a key factor contributing to low vaccination rates in children, along with misconceptions such as the perception of seasonal influenza as a mild illness, concerns about the potential harm of the influenza vaccine, and the belief that vaccination is unnecessary. These results are consistent with other studies’ findings [17,18]. Moreover, insufficient and incorrect understanding of the severe outcomes of seasonal influenza may lead to ambiguous attitudes and practices regarding vaccination. Various factors, such as the absence of recommendations from healthcare providers, concerns over potential vaccine side effects, and doubts about its efficacy, contribute to vaccine refusal [14,28]. To improve vaccine uptake, addressing these misconceptions through media campaigns, awareness initiatives, and healthcare professionals is crucial.

Physicians and healthcare professionals are pivotal in raising awareness and dispelling misconceptions about seasonal influenza and its vaccine. However, in this study, only 17.8% of participants reported receiving information from medical staff, with the majority obtaining knowledge through social media. This contrasts with findings from prior research [12,14,17,18,28]. To address this, the Saudi Ministry of Health and hospital administrations should ensure that healthcare professionals are well equipped with updated knowledge about influenza complications and the importance of vaccination. Additionally, leveraging social media’s widespread influence in health campaigns can enhance public awareness and positively shape attitudes toward vaccination.

In our study, the majority of parents displayed a favorable stance toward seasonal influenza vaccination, echoing the results of Alolayan et al. [17]. However, some participants’ negative attitudes stemmed from limited understanding of the vaccine’s safety and effectiveness. A study conducted in Riyadh revealed that over 90% of participants trusted the Saudi Ministry of Health and relied on information from physicians [12]. In a similar vein, research in Jordan demonstrated that providing clear and accurate information about the vaccine’s benefits significantly shaped positive parental attitudes toward vaccination [28].

Our study found no significant difference between males and females in their attitudes toward seasonal influenza vaccination, aligning with earlier findings by Alolayan et al. [17]. However, this contrasts with the results of Alenazi, which may be attributed to the higher proportion of working males in Riyadh, where their study was conducted [18]. Employment among males in Riyadh could contribute to increased knowledge and awareness, fostering a more positive attitude. Interestingly, in our study, medical professionals did not demonstrate superior knowledge or more optimistic attitudes toward vaccination, despite their pivotal role as disseminators of accurate information about the seasonal influenza vaccine in Saudi Arabia. This observation aligns with other studies indicating that employment in the medical field does not necessarily ensure better knowledge or attitudes regarding immunization [30,31]. These discrepancies highlight the complexity of factors influencing vaccination attitudes across different studies.

Previous research has established a significant correlation between parents’ seasonal influenza vaccination status and the likelihood of their children receiving the vaccine. These studies have shown that children are more likely to be vaccinated when their parents are immunized [17,28]. Our findings support this connection, highlighting the crucial role of enhancing parental knowledge and attitudes to improve vaccination rates among children. As expected, parents who were informed about the vaccine, who vaccinated themselves and their children, and who demonstrated a strong understanding and positive attitude toward the seasonal influenza immunization program were key contributors to the higher vaccination uptake.

Our study was not free of limitations: (1) it was conducted in a single region of Saudi Arabia, limiting generalizability; (2) the cross-sectional design restricts causal inference; (3) self-reported health conditions may introduce response bias; and (4) convenience sampling could lead to sampling bias. To our knowledge, we are considered the first study in Al-Madinah, offering valuable insights into local SIV attitudes and behaviors. Additionally, the large sample size and findings contribute significantly to understanding vaccination uptake. Our results highlight the need for educational programs to enhance parental awareness of the vaccine’s importance. Emphasizing the consequences of vaccine refusal through social media and public campaigns could strengthen parental responsibility for their children’s health.

## 5. Conclusions

While most parents recognize the importance of vaccination, misconceptions about vaccine safety and perceived low need remain prevalent, alongside barriers such as vaccine availability. Sociodemographic factors such as education, income, and profession were linked to better knowledge and positive attitudes. Given these findings, future research should focus on overcoming misconceptions and enhancing access to the vaccine, particularly through targeted educational campaigns and social media outreach. Further studies in diverse regions and longitudinal designs are recommended to explore causality and assess the effectiveness of intervention strategies.

## Figures and Tables

**Table 1 ijerph-22-01704-t001:** Descriptive analysis of the parents’ sociodemographic characteristics. N = 407.

	Frequency	Percentage
**Sex**		
Male	177	43.5
Female	230	56.5
**Age (years)**		
18–24 years	61	15
25–34 years	163	40
35–44 years	139	34.2
45–54 years	29	7.1
≥55 years	15	3.7
**Nationality**		
Non-Saudi	54	13.3
Saudi	353	86.7
**Educational Level**		
Less than high school	41	10.1
High school	123	30.2
College/university degree	208	51.1
Higher studies	35	8.6
**Household Monthly Income**		
Don’t know/prefer not to answer	132	32.4
SAR 2000–5999	100	24.6
SAR 6000–9999	94	23.1
SAR 10,000–15,000	63	15.5
SAR > 15,000	18	4.4
**Medical Professional**		
No	328	80.6
Yes	79	19.4
**Number of Children, Mean (SD)**		2.40 (1.5), median = 2

**Table 2 ijerph-22-01704-t002:** Descriptive analysis of the parents’ awareness of and behavior toward the children’s seasonal flu vaccine.

	Frequency	Percentage
**Have you ever heard about the seasonal influenza vaccine before?**		
No	18	4.4
Yes	389	95.6
**Where did you hear/learn about the seasonal influenza vaccine?**		
From medical staff	63	17.8
Media (like TV/social media channels)	142	40.1
From family or friends	59	16.7
From internet sources	20	5.6
Awareness campaign	70	19.8
**Did you receive the influenza vaccine as a parent?**		
No	164	40.3
Yes	243	59.7
**Is your child up to date with the national vaccination program?**		
No	92	22.6
Yes	295	72.5
Unsure	20	4.9
**Did you/your child receive the flu vaccine before?**		
No	163	40
I don’t know	63	15.5
Yes	181	44.5
**If you had never had a seasonal flu vaccine, why? *n* = 128**		
I think it is harmful	21	16.4
It is not available	30	23.4
Influenza is simple	46	35.9
No need to vaccinate my child	46	35.9
I think it is not effective	33	25.8
I believe it can cause influenza signs and symptoms	7	5.5
**Have you been practicing infection control measures to protect you from contracting COVID-19 at crowded areas (masks, hand hygiene, social distancing)?**		
Never	6	1.5
Seldom/rarely	14	3.4
Sometimes	70	17.2
Usually/often	111	27.3
Always/very often	206	50.6
**Have you been infected with COVID-19 (either you or any family member)?**		
No	197	48.4
Yes	210	51.6
**Are you willing to take the seasonal influenza vaccine this year?**		
No	130	31.9
Yes	277	68.1

**Table 3 ijerph-22-01704-t003:** Descriptive analysis of the parents’ knowledge of the children’s seasonal flu vaccine.

	Incorrect Answer *n* (%)	Correct Answer *n* (%)
Influenza illness is a common disease?	93 (22.9)	314 (77.1)
Flu infection is highly contagious?	145 (35.6)	262 (64.4)
Influenza illness transmitted through coughing and sneezing?	96 (23.6)	311 (76.4)
Influenza can lead to child hospitalization and even to death?	213 (52.3)	194 (47.7)
Influenza vaccine is recommended to all children more than 6 months?	162 (39.8)	245 (60.2)
The aim of the influenza vaccine is to make the flu less severe?	221 (54.3)	186 (45.7)
Flu vaccine could cause influenza?	313 (76.9)	94 (23.1)
Influenza vaccine should be given every year?	121 (29.7)	286 (70.3)
Antibiotics can treat influenza virus infection?	316 (77.6)	91 (22.4)
Seasonal influenza vaccine is available in the hospital and PHCs?	114 (28)	293 (72)
Which of the following is a seasonal flu sign and/or symptom?		
Fever	64 (15.7)	343 (84.3)
Runny nose	142 (34.9)	265 (65.1)
Cough	230 (56.5)	177 (43.6)
Vomiting	310 (76.2)	97 (23.8)
Diarrhea	319 (78.4)	88 (21.6)
Weakness	224 (55)	183 (45)

**Table 4 ijerph-22-01704-t004:** Descriptive bivariate analysis of the parents’ knowledge of the flu vaccine across levels of their sociodemographic characteristics and other perceptions. N = 407.

	Mean (SD) Total Knowledge Score out of 16	*p*-Value
**Sex**		
Male	8.24 (3.14)	0.297
Female	8.57 (3.04)	
**Age (years)**		**0.001**
18–24 years	7.72 (3.16)	
25–34 years	8.52 (3.10)	
35–44 years	8.88 (2.79)	
45–54 years	8.66 (3.21)	
≥55 years	5.67 (3.50)	
**Nationality**		
Non-Saudi	8.74 (3.15)	0.420
Saudi	8.38 (3.10)	
**Educational Level**		
Less than high school	6.49 (2.78)	**<0.001**
High school	8.29 (3.30)	
College/university degree	8.83 (2.91)	
Higher studies	8.80 (2.84)	
**Household Monthly Income**		
Don’t know/prefer not to answer	7.86 (3.20)	**<0.001**
SAR 2000–5999	7.71 (2.98)	
SAR 6000–9999	8.78 (2.81)	
SAR 10,000–15,000	9.979 (3.02)	
SAR > 15,000	9.89 (2.14)	
**Medical Professional**		
No	8.16 (3.05)	**<0.001**
Yes	9.53 (2.99)	
**Have you ever heard about the seasonal influenza vaccine before?**		
No	3.17 (2.04)	**<0.001**
Yes	8.66 (2.90)	
**Did you receive influenza vaccine as a parent?**		
No	7.35 (3.20)	**<0.001**
Yes	9.15 (2.78)	
**Is your child up to date with the national vaccination program?**		
No	6.83 (3.50)	**<0.001**
Yes	8.96 (2.76)	
Unsure	7.85 (3.12)	
**Did you/your child receive flu vaccine before?**		
No	8.15 (2.97)	**<0.001**
I don’t know	6.51 (3.26)	
Yes	9.34 (2.77)	
**Have you been infected with COVID-19 (either you or any family member)?**		
No	8.24 (3.29)	0.253
Yes	8.60 (2.88)	
**Are you willing to take the seasonal influenza vaccine this year?**		
No	8.21 (3.16)	0.331
Yes	8.53 (3.05)	

All significant *p*-values were demonstrated in bold.

**Table 5 ijerph-22-01704-t005:** (a) Descriptive analysis of the parents’ knowledge of the children’s seasonal flu vaccine. (b) Descriptive statistics of the parents’ overall perceptions about seasonal influenza vaccination and disease.

**(a)**			
	**Mean (SD)**	**Disagree *n*** **(%)**	**Agree *n*** **(%)**
Vaccinations are important to keep children healthy.	3.53 (0.60)	17 (4.1)	390 (95.8)
The seasonal influenza vaccine is necessary for my children.	3.10 (0.78)	93 (22.8)	314 (77.2)
The seasonal influenza vaccine is a safe vaccine.	3.11 (0.72)	77 (18.9)	330 (81.1)
The seasonal influenza vaccine is an effective vaccine.	3.01 (0.76)	99 (24.4)	308 (75.7)
Influenza illness can lead to serious complications.	2.90 (0.92)	126 (31)	281 (69)
I trust the information given to me by the Saudi Ministry of Health.	3.51 (0.60)	18 (4.4)	389 (95.6)
I trust the information given by my doctor.	3.47 (0.61)	23 (5.6)	384 (94.4)
**(b)**			
	**Mean (SD)**	**Maximum Possible Score**	
Overall attitude toward influenza vaccines.	22.48 (3.39)	7–35 points	
Knowledge of influenza symptoms.	2.84 (1.60)	0–6 points	
Knowledge of the influenza disease.	5.59 (2.30)	0–10 points	
Overall total influenza knowledge score.	8.43 (3.10)	0–16 points	

**Table 6 ijerph-22-01704-t006:** Descriptive bivariate analysis of the parents’ attitudes toward the flu vaccine across levels of their sociodemographic characteristics and other perceptions. N = 407.

	Mean (SD) Attitude Toward Vaccine (out of 35)	*p*-Value
**Sex**		
Male	22.13 (350)	0.071
Female	22.74 (3.30)	
**Age (years)**		
18–24 years	21.70 (3.20)	0.255
25–34 years	22.64 (3.35)	
35–44 years	22.43 (3.31)	
45–54 years	22.83 (4.32)	
≥55 years	23.53 (3.41)	
**Nationality**		
Non-Saudi	22.54 (3.68)	0.889
Saudi	22.50 (3.36)	
**Educational Level**		
Less than high school	22.41 (3.24)	0.248
High school	22.04 (3.12)	
College/university degree	22.63 (3.51)	
Higher studies	23.20 (3.74)	
**Household Monthly Income**		
Don’t know/prefer not to answer	22.00 (3.14)	0.273
SAR 2000–5999	22.40 (3.28)	
SAR 6000–9999	22.87 (8.68)	
SAR 10,000–15,000	22.92 (3.60)	
SAR > 15,000	22.78 (3.41)	
**Medical Professional**		
No	22.39 (3.28)	0.314
Yes	22.82 (3.85)	
**Have you ever heard about the seasonal influenza vaccine before?**		
No	21.72 (4.42)	0.336
Yes	22.51 (3.34)	
**Did you receive the influenza vaccine as a parent?**		
No	21.65 (3.42)	**<0.001**
Yes	23.04 (3.27)	
**Is your child up to date with the national vaccination program?**		
No	21.71 (3.55)	**0.037**
Yes	22.74 (3.34)	
Unsure	22.20 (3.11)	
**Did you/your child receive the flu vaccine before?**		
No	21.80 (3.41)	**0.001**
I don’t know	22.38 (3.13)	
Yes	23.12 (3.37)	
**Have you been infected with COVID-19 (either you or any family member** **)?**		
No	22.20 (3.42)	0.116
Yes	22.73 (3.36)	
**Are you willing to take the seasonal influenza vaccine this year?**		
No	21.36 (3.74)	**<0.001**
Yes	23.00 (3.10)	

All significant *p*-values were demonstrated in bold.

**Table 7 ijerph-22-01704-t007:** (a) Multivariable linear regression analysis of people’s mean knowledge of flu disease. (b) Multivariable linear regression analysis of people’s mean perceived attitude toward flu vaccination.

**(a)**				
		**95% CI for Beta**		
	**Unstandardized Beta** **Coefficient**	**Lower**	**Upper**	** *p* ** **-Value**
(Constant)	1.342	−0.915	3.598	0.243
Sex = Female	0.250	−0.294	0.794	0.367
Age Group: ≥55 Years vs. Younger Age Groups	−0.372	−0.684	−0.060	0.020
Educational Level	0.187	−0.176	0.550	0.311
Household Monthly Income SAR ≥10,000/Month	0.418	0.168	0.667	0.001
Number of Children in the Family	0.008	−0.197	0.214	0.937
Had Received Influenza Vaccine as Parent = Yes vs. No	1.132	0.572	1.693	0.000
Job: Medical Professional	0.709	0.000	1.418	0.050
Mean Perceived Attitudes toward Pediatric Seasonal Flu Vaccine Score	0.053	−0.027	0.133	0.196
Prior Awareness of Seasonal Flu Vaccine: Yes	4.185	2.853	5.518	0.000
Own Children Are Up to Date with the National Vaccination Program: Yes	0.488	0.176	0.801	0.002
**(b)**				
		**95% CI for Beta**		
	**Unstandardized Beta** **Coefficient**	**Lower**	**Upper**	** *p* ** **-Value**
(Constant)	20.332	18.827	21.837	<0.001
Sex = Female vs. Male	0.405	−0.224	1.034	0.207
Age Group	0.334	−0.026	0.693	0.069
Educational Level	0.095	−0.322	0.511	0.655
Household Monthly Income	0.084	−0.209	0.376	0.574
Number of Children	−0.376	−0.610	−0.143	0.002
Had Received Influenza Vaccine as Parent = Yes vs. No	0.995	0.346	1.644	0.003
Mean Knowledge of Flu Symptoms and Signs Total Score	−0.522	−0.720	−0.325	<0.001
Mean General Knowledge of Flu Disease Score	0.461	0.314	0.608	<0.001
Job: Medical Professional	−0.284	−1.108	0.541	0.499

## Data Availability

All data are included in this manuscript.

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
