# Peer review of "Parental Knowledge, Awareness, and Attitudes Toward Seasonal Influenza Vaccination in Al-Madinah, Saudi Arabia: A Cross-Sectional Study"

_ijerph, 2025, doi:10.3390/ijerph22111704_

Round 1
Reviewer 1 Report
Comments and Suggestions for Authors
The topic of this paper is relevant and important for public health, as parents’ perceptions strongly influence children’s influenza vaccination rates. Although similar studies have been conducted, this paper provides useful local data from Al-Madinah.
We found there are some methodological and analytical limitations that weaken the overall quality of the study. The use of convenience sampling may cause bias and affect the generalizability of the results, and this should be clearly discussed by the authors. The statistical analysis is also quite simple and lacks multivariate regression, which would help identify independent factors influencing parental knowledge and attitudes.
The discussion section is brief and mostly repeats the results instead of comparing them with findings from previous Knowledge–Attitude–Practice (KAP) studies. It should be expanded with more interpretation and literature support. In addition, the reference style and formatting need to be revised to meet IJERPH standards.
Overall, the topic is valuable, but the manuscript needs more analytical depth and methodological improvement before it can be considered for publication.
- Discuss the implications of using convenience sampling and its impact on generalizability.
- Incorporate multivariate analysis to better assess predictors of knowledge and attitudes.
- Deepen the discussion by referencing prior KAP studies and comparing findings.
- Revise the writing style and improve consistency in references and formatting.
Author Response
|
Reviewer |
Comment |
Response |
|
Reviewer #1 |
Discuss the implications of using convenience sampling and its impact on generalizability. |
The manuscript has been revised to: reflect that the convenient sample was used inside each primary health care center. However, the participating centers were distributed across Madinah city to ensure representation from different communities. The method section was edited. |
|
Reviewer #1 |
Incorporate multivariate analysis to better assess predictors of knowledge and attitudes. |
The manuscript has been revised to: add the multivariate regression analysis. Both method and result section were edited. |
|
Reviewer #1 |
Deepen the discussion by referencing prior KAP studies and comparing findings. . |
Thank you for your comment. Previous studies are already included in the discussion. |
|
Reviewer #1 |
Revise the writing style and improve consistency in references and formatting. . |
Edited |
Reviewer 2 Report
Comments and Suggestions for Authors
Journal: IJERPH (ISSN 1660-4601)
Manuscript ID: ijerph-3927585
Type: Article
Title: Parental Knowledge, Awareness, and Attitudes Toward Seasonal Influenza Vaccination in Al-Madinah, Saudi Arabia: A Cross-Sectional Study
Authors: Abdulsalam Alawfi * , Muhammad Tobaiqi , Osama Algrigri , Amal H. Aljohani , Amal Mohammed Q. Surrati , Bandar Albaradi , Amer Alshengeti
A brief summary
Dear Authors, I wish to express my gratitude for the opportunity to review your excellently written manuscript. The topic you address is both timely and engaging. The prevention of all diseases is a cornerstone of sound public health, particularly when such prevention can be achieved through a straightforward method such as targeted vaccination of the target population. Only minor revisions to the manuscript are required, and my specific comments are provided below:
Line 20 and Line 89: Please clarify and specify the exact age (for example, from 6 months to 18 years).
Participants and Recruitment (Line 103-106): Kindly clarify the mode of conducting the interviews—whether they were administered in written or oral form. Additionally, indicate the time period during which this cross-sectional study was conducted.
Data collection (Line 107-114): Please clarify the language and population in which the referenced questionnaire was validated, as well as the language used for your questionnaire in the present study. Furthermore, are there any more detailed data on the validation process itself, such as Cronbach's alpha?
There are several repeated typographical errors at Lines 153, 169, 178, 193, and 202: The punctuation mark should be placed after the table reference, e.g., (23.4%) Table 2.
I recommend separating the text on study limitations from the discussion by adding the subheading “Study limitations.” Additionally, it would be beneficial to include the subheading “Public health implications and recommendations for future research.”
All tables are clear and well-structured; no changes are necessary. The introduction and discussion are well-organized and logically presented. The conclusion clearly supports and aligns with the results obtained.
Lines 339-415: All references should be carefully reviewed to ensure full compliance with the journal’s author guidelines.
The English is fine.
Author Response
|
Reviewer #2 |
Line 20 and Line 89: Please clarify and specify the exact age (for example, from 6 months to 18 years). |
Added and edited |
|
Reviewer #2 |
Participants and Recruitment (Line 103-106): Kindly clarify the mode of conducting the interviews—whether they were administered in written or oral form. Additionally, indicate the time period during which this cross-sectional study was conducted |
Clarified under Data collection section |
|
Reviewer #2 |
Data collection (Line 107-114): Please clarify the language and population in which the referenced questionnaire was validated, as well as the language used for your questionnaire in the present study. Furthermore, are there any more detailed data on the validation process itself, such as Cronbach's alpha? |
The survey was translated from English to Arabic language to facilitate the interview by the data collector team members who are bilingual. Ten interviews were done before the start of study to ensure clarity of the content. Method section has been edited |
|
Reviewer #2 |
There are several repeated typographical errors at Lines 153, 169, 178, 193, and 202: The punctuation mark should be placed after the table reference, e.g., (23.4%) Table 2 |
typographical errors at these lines were corrected and punctuation marks were edited |
|
Reviewer #2 |
I recommend separating the text on study limitations from the discussion by adding the subheading “Study limitations.” Additionally, it would be beneficial to include the subheading “Public health implications and recommendations for future research.” |
Thank you for this great suggestion . however we kept study limitations and Public health implications included in discussion section to be compliant with journal’s author guidelines |
|
Reviewer #2 |
Lines 339-415: All references should be carefully reviewed to ensure full compliance with the journal’s author guidelines. |
All references were reviewed and should be compliant with journal’s author guidelines. |
Reviewer 3 Report
Comments and Suggestions for Authors
- A brief summary A cross-sectional survey about influenza vaccination of children in Saudi Arabia from the perspective of the parents.
- Comments regarding general concepts: Not a longitudinal study, just enought participants, besides the CDC & WHO & ACIP recommendations also e.g. European CDC (ECDC) recommendations should be included because the influena vaccination of children is not an universal practice.
- Reviews: it is relevant for Saudi Arabia, but it not necessarily identified a gap in knowledge about influenza vaccination outside Saudi Arabia, the references are often too old (often more than 5 years old, sometimes even older than 5 years)
- Specific comments: p value if significant must presented in bolt to make it clear immediately
The manuscript clear, relevant to the field of influenza vaccination, and presented in a well-structured manner.
A lot of the references are older than 5 years. No self-citations were found.
The manuscript scientifically sound, and the experimental design is appropriate to test the hypothesis.
The manuscript’s results are reproducible based on the details given in the Methods section.
The tables are appropriate and present the date properly. The data are interpreted appropriately and consistently throughout the manuscript.
The conclusions are consistent with the evidence and arguments presented.
Author Response
|
Reviewer #3 |
p value if significant must presented in bolt to make it clear immediately |
All significant p values are presented in bolt |
|
Reviewer #3 |
A lot of the references are older than 5 years. No self-citations were found |
We appreciate this comment. Howvere, as most reserches in this area are coming from developed countries where the issue of influenza vaccine have addressed years ago. We could not find more recent studies addressing same topic. |
|
Reviewer #3 |
Not a longitudinal study, just enought participants, besides the CDC & WHO & ACIP recommendations also e.g. European CDC (ECDC) recommendations should be included because the influena vaccination of children is not an universal practice. |
Thak you for this comment. This has been mentioned in the introduction line 61-65. Europian CDC has the same recommendation as WHO. |
Round 2
Reviewer 1 Report
Comments and Suggestions for Authors
The Authors have revised the manuscript well.